# Bridging the Gap between Object and Image-level Representations for Open-Vocabulary Detection

**Hanoona Rasheed[1,*], Muhammad Maaz[1,*], Muhammad Uzair Khattak[1],**
**Salman Khan[1,2], Fahad Shahbaz Khan[1,3]**
[1]Mohamed bin Zayed University of AI, UAE
[2]Australian National University, Australia    [3]Linköping University, Sweden

## Abstract

Existing open-vocabulary object detectors typically enlarge their vocabulary sizes by leveraging different forms of weak supervision. This helps generalize to novel objects at inference. Two popular forms of weak-supervision used in open-vocabulary detection (OVD) include pretrained CLIP model and image-level supervision. We note that both these modes of supervision are *not* optimally aligned for the detection task: CLIP is trained with image-text pairs and lacks precise localization of objects while the image-level supervision has been used with heuristics that do not accurately specify local object regions. In this work, we propose to address this problem by performing object-centric alignment of the language embeddings from the CLIP model. Furthermore, we visually ground the objects with only image-level supervision using a pseudo-labeling process that provides high-quality object proposals and helps expand the vocabulary during training. We establish a bridge between the above two object-alignment strategies via a novel weight transfer function that aggregates their complimentary strengths. In essence, the proposed model seeks to minimize the gap between object and image-centric representations in the OVD setting. On the COCO benchmark, our proposed approach achieves 36.6 $AP_{50}$ on novel classes, an absolute 8.2 gain over the previous best performance. For LVIS, we surpass the state-of-the-art ViLD model by 5.0 mask AP for rare categories and 3.4 overall. Code: `https://github.com/hanoonaR/object-centric-ovd`.

## 1   Introduction

Open-vocabulary detection (OVD) aims to generalize beyond the limited number of base classes labeled during the training phase. The goal is to detect novel classes defined by an unbounded (open) vocabulary at inference. Owing to the challenging nature of the OVD task, different forms of weak-supervision for novel categories are typically used, *e.g.*, extra image-caption pairs to enlarge the vocabulary [1], image-level labels on classification datasets [2] and pretrained open-vocabulary classification models like CLIP [3]. The use of weak-supervision to enlarge the vocabulary is intuitive as the cost of annotating large-category detection datasets is monumental while the image-text/label pairs are readily available via large classification datasets [4] or internet sources [3, 5].

One of the major challenges with enlarging vocabulary via image-level supervision (ILS) or pretrained models learned using ILS is the inherent mis-match between region and image-level cues. For instance, pretrained CLIP embeddings used in the existing OVD models [6, 2] do not perform well in locating object regions [7] since the CLIP model is trained with full scale images. Similarly, weak supervision on images using caption descriptions or image-level labels does not convey the precise object-centric information. For label grounding in images, the recent literature explores expensive pretraining with auxiliary objectives [1] or use heuristics such as, the max-score or max-size boxes [2].

---

[*]Equal contribution

36th Conference on Neural Information Processing Systems (NeurIPS 2022).

In this paper, we set out to bridge the gap between object and image-centric representations within the OVD pipeline. To this end, we propose to utilize high-quality class-agnostic and class-specific object proposals via the pretrained multi-modal vision transformer (ViT) [8]. The *class-agnostic* object proposals are then used to distill region-specific information in the CLIP visual embeddings, making them suitable for local objects. Furthermore, the *class-specific* proposal set allows us to visually ground a larger vocabulary, thereby aiding in generalization to novel categories. Next, the final and important question is how to make visual-language (VL) mapping amenable to local object-centric information. For this purpose, we introduce a region-conditioned weight transfer process which closely ties together image and region VL mapping. In a nut-shell, the proposed approach connects the image, region and language representations to generalize better to novel open-vocabulary objects.

The major contributions of this work include:

- We propose *region-based knowledge distillation* to adapt image-centric CLIP embeddings for local regions, thereby improving alignment between region and language embeddings. We show that the resulting well-aligned representations aid in improving the overall performance of our text driven OVD pipeline.

- In order to visually ground weak image labels, our approach performs *pseudo-labeling* using the high-quality object proposals from pretrained multi-modal ViTs. This helps in enlarging the class vocabulary and therefore generalizes better to new object classes.

- The above contributions mainly target the visual domain. In order to preserve the benefits of object-centric alignment in the language domain, we also propose to explicitly condition the (pseudo-labeled) image-level VL mapping on the region-level VL mapping via a novel *weight transfer function*. In this manner, we are the first to simultaneously integrate object-centric visual and language alignment within a single architecture for OVD.

- Our extensive experiments demonstrate the improved OVD capability of the proposed approach. On COCO and LVIS benchmarks, our method achieves absolute gains of 8.2 and 5.0 AP on novel and rare classes over the current SOTA methods. Further generalizability is demonstrated by our cross-dataset evaluations performed on COCO, OpenImages and Objects365, leading to consistent improvements compared to existing methods.

## 2 Related Work

**Zero-shot Object Detection (ZSD):** This setting involves detecting novel class objects at inference, for which no visual examples are available during training. Zhu *et al.* [9] use semantic information with visual features to get proposals for both seen and unseen classes. Bensal *et al.* [10] show that learning a good separation between background and foreground is critical in ZSD and propose to use multiple latent classes for modeling background during training. Rahman *et al.* [11] propose a polarity loss to solve the ambiguity between background and unseen classes. DELO [12] focuses on generating good proposals for unseen classes by synthesizing visual features for unseen objects using a generative model. Gupta *et al.* [13] benefits from the contemporary cues in semantic and visual space ensuring better class separation for ZSD. Other works use additional learning signals, including unlabeled images from target domain [14] and raw textual descriptions from the internet [15]. Although significant progress has been made on this topic [14, 15, 13], the inherent complexity of the task makes it challenging for the ZSD models to generalize well to unseen object classes.

**Weakly-supervised Object Detection (WSOD):** In this setting, only image-level labels are used to approach object detection [16, 17, 18, 19, 20], or are used alongside the detection dataset to enlarge the detector vocabulary [21, 22, 23]. Bilen *et al.* [24] proposed a weakly-supervised deep detection network (WSDNN) that uses off-the-shelf region proposals [25, 26] and computes objectness and recognition scores for each proposal using separate subnetworks. Cap2Det [27] operates in a similar setting and uses raw text captions to generate pseudo-labels to guide image-level supervision. Li *et al.* [28] uses segmentation-detection collaborative network (SDCN) for accurate detection under weakly-supervised setting using only image labels. PCL [29] proposes to cluster the spatially adjacent proposals and then assign image labels to each cluster. CASD [30] argues that the detectors trained only with image-level labels are prone to detect boxes around salient objects and propose feature attention along with self-distillation to address the issue. YOLO9000 [31] and DLWL [32] augments the detection training by assigning image-level labels to the max-score proposal. Detic [2] shows that using max-size proposal is an optimal choice for assigning image-level labels as it does not rely on the predictions of the network being optimized and provides better signals for the novel classes.

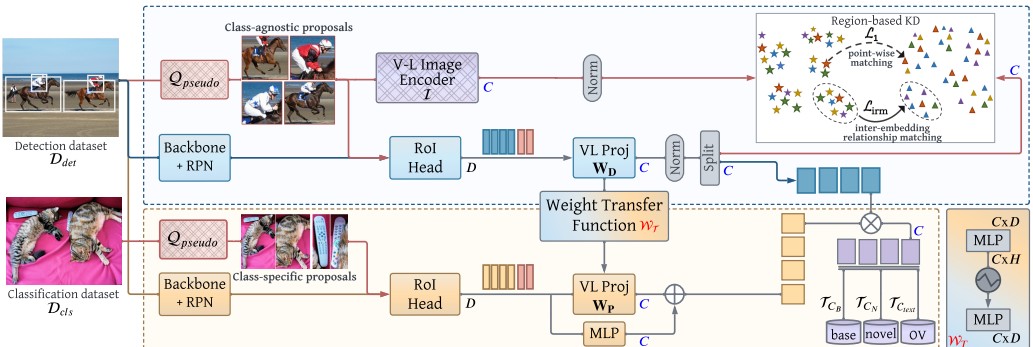

Figure 1: **An overview of our proposed object-centric framework for OVD.** We pair a two-stage object detector with fixed language embeddings from a pretrained visual-language (VL) model, CLIP [3]. Our proposed pseudo-labeling strategy $\mathcal{Q}_{\text{pseudo}}$ uses pretrained multi-modal ViTs to obtain high-quality class-agnostic and class-specific proposals. The overall pipeline follows a stage-wise learning strategy. *First,* we introduce region-based knowledge distillation (RKD) to adapt image-centric CLIP embeddings for local regions. Using the pretrained VL image encoder as a teacher model, we train the detector to induce point-wise and inter-embedding relationship alignment with our region embeddings using class-agnostic proposals from $\mathcal{Q}_{\text{pseudo}}$. *Next,* we utilize a weakly-supervised learning framework by combining instance-level labels from detection dataset and image-level labels from classification dataset which are visually grounded using $\mathcal{Q}_{\text{pseudo}}$. This weak-supervision helps in enlarging the class vocabulary and generalizes the detector to novel classes. To preserve the benefits of object-centric alignment in the language domain learned via RKD, we explicitly condition the image-level VL mapping $W_P$, on the learned region-level VL mapping $W_D$ via a novel weight transfer function.

We also operate in a similar WSOD setting and use high-quality object proposals from pretrained multi-modal ViT [8] to enlarge detector vocabulary and generalize towards novel object categories.

**Open-vocabulary Object Detection (OVD):** In OVD, the objective is to detect target class objects not present in the training/base class vocabulary. A typical solution of the problem is to replace the classifier weights with text embeddings of the target vocabulary (*e.g.*, GloVe [33], BERT [34], CLIP [3]). OVR-RCNN [1] uses BERT embeddings as classifier weights and proposes to use open-vocabulary captions to learn the vision-to-language mapping. It surpasses the ZSD approaches by a large margin. ViLD [6] uses pretrained CLIP [3] to distill knowledge into a two-stage object detector [35] and replaces the classifier weights with CLIP text embeddings obtained by ensembling multiple text prompts (*e.g.*, a {category}, a photo of a {category}). Gao *et al.* [36] generate pseudo bounding-box labels using pretrained VL models for training open-vocabulary detector. All these methods use carefully designed manual prompts for generating text embeddings. DetPro [37] and PromptDet [38] replace these manual prompts with learnable tokens and achieve competitive results on novel/rare categories. However, in our work, we use fixed manual prompts and instead focus on improving the object-centric representations for open-vocabulary object detection.

## 3    Object-centric Open-Vocabulary Detection

Here, we first present a brief overview of the proposed open-vocabulary detection (OVD) framework. As discussed earlier, existing OVD methods use different forms of weak supervision that employ image-centric representations, making them less suited for the end detection task. Our proposed method aims to bridge the gap between image and object-centric visual-language (VL) representations. We summarize the architectural overview of our method in Fig. 1. The proposed design has three main elements. 1) Our *region-based knowledge distillation* (refer Sec. 3.2) adapts image-centric language representations to be object-centric. A VL mapping learns to align the local region representations of the detector to the language representations by distilling the detector's region representations with region representations from a VL model (CLIP). 2) Given weak image-level supervision, we use *pseudo-labeling* from pretrained multi-modal ViTs (refer Sec. 3.3) to improve generalization of the detector to novel classes. 3) For an efficient combination of the above two proposed components, we condition the VL mapping learned during the weak supervision on the VL mapping learned with region-based distillation via a novel *weight transfer function* (refer Sec. 3.4). Specifically, we follow a stage-wise learning strategy to first align the region and language embeddings using RKD, and use this distilled VL mapping for object-centric visual and language alignment in the subsequent stage.

## 3.1 Detection Pipeline: Preliminaries

In the open-vocabulary detection problem, we have access to an object detection dataset where the training set, $\mathcal{D}_{\text{det}}$, comprises samples from the set of base object categories, $\mathcal{C}_{\text{B}}$. The images of $\mathcal{D}_{\text{det}}$ are exhaustively annotated with bounding-box labels and corresponding class labels $y_r \in \mathcal{C}_{\text{B}}$, for the different objects in the image. Given an image $I \in \mathbb{R}^{H \times W \times 3}$, we design an open-vocabulary object detector to solve two subsequent problems: (1) effectively localize all objects in the image, (2) classify the detected region into one of the class label of $\mathcal{C}_{\text{test}}$, which is provided by the user at test time. The categories during test time also include novel categories $\mathcal{C}_{\text{N}}$ beyond the closed set of base categories seen during the training phase, *i.e.*, $\mathcal{C}_{\text{test}} = \mathcal{C}_{\text{B}} \cup \mathcal{C}_{\text{N}}$.

We convert a generic two-stage object detector [35] to an open-vocabulary detector by replacing the learnable classifier head with fixed language embeddings, $\mathcal{T}$ corresponding to the category names of $\mathcal{C}_{\text{test}}$, that are obtained using a large-scale pretrained VL model. Following [6], we use the *text embeddings* from CLIP text encoder [3] for classification, where only the embeddings of $\mathcal{C}_{\text{B}}$ categories, $\mathcal{T}_{\mathcal{C}_{\text{B}}}$ are used during training. Specifically, we generate the text embeddings offline, by processing the prompts corresponding to each category with a template of 'a photo of {category}' through the CLIP text encoder. The RoI [35] head computes pooled feature representations $\phi(r)$ of the proposals $r$ generated by the region proposal network (RPN). These feature embeddings are projected to a common feature space shared by the text embedding $\mathcal{T}$ using a linear layer $f(\cdot)$, which we represent as *region embeddings*, $\mathcal{R} = f(\phi(r)) \in \mathbb{R}^D$. For classification, we compute the cosine similarity between the region embeddings and text embeddings to find the matching pairs. During training, the regions that do not match with any of the ground-truths are assigned to the background category represented by a fixed all zero embedding. We compute the cosine similarity by comparing each region to each base class, $\mathcal{V} = sim(r, b) = \cos\left(\mathcal{R}(r), \mathcal{T}_b\right) \forall b \in \mathcal{C}_{\text{B}}$. The classification loss is a softmax cross-entropy (CE) where the logits are the cosine similarity scores,

$$\mathcal{L}_{cls} = \frac{1}{N} \sum_r \mathcal{L}_{CE}\left(\text{softmax}\left(\frac{\mathcal{V}}{\tau}\right), y_r\right), \;\; y_r \in \mathcal{C}_{\text{B}}.$$

where $\tau$ is the temperature, $N$ is the total number of proposals per image, and $r$ represents a single proposal with the ground-truth label $y_r$.

## 3.2 Region-based Knowledge Distillation

In the OVD setting, we assume that $f(\cdot)$ learns a VL mapping and aligns the output region embeddings of the detector with the corresponding CLIP text embeddings. However, the performance on novel categories is not comparable to what CLIP encoded embeddings would provide (refer Appendix B for details). We hypothesize that this performance gap is mainly due to two reasons, i) the data that has been used for training CLIP model consist of scene-centric images, making it less suitable for region classification, *e.g.,* in our case where object-centric tightly bounded proposals are used, ii) the zero-shot generalization ability of the pair-wise trained CLIP image and text embeddings cannot be fully utilized due to the mismatch between regions representations from CLIP image encoder and our detector. Based on these insights, we propose a ***region-based knowledge distillation*** (**RKD**).

The proposed RKD uses distillation in the detection pipeline by distilling region embeddings from high-quality class-agnostic proposals ($\tilde{r}$) obtained from a pretrained multi-modal ViT (MViT) [8]. Note that we obtain both class-agnostic (used in RKD) and class-specific (refer Sec. 3.3) object proposals using this pseudo-labeling process, which we refer to as $\mathcal{Q}_{\text{pseudo}}$. This is possible via using intuitive text queries to interact with the MViT model that can locate generic objects and provides the corresponding set of candidate proposals. The queries can be generic or targeted, based on the task, *e.g.*, 'all objects' to generate class-agnostic proposals, or 'every dog' for a specific class.

For RKD, we compute class agnostic proposals on $\mathcal{D}_{\text{det}}$ using simple text query, 'all objects' and select top-K proposals (Fig. 3b). CLIP embeddings $\mathcal{I}(\tilde{r})$ are then computed offline using the CLIP image encoder $\mathcal{I}(\cdot)$. With the detector region embeddings and the corresponding CLIP region representations, we propose to use two types of distillation losses to improve the alignment.

**(1) *Point-wise embedding matching loss:*** The $\mathcal{L}_1$ loss matches the individual region embeddings $\tilde{\mathcal{R}} = f(\phi(\tilde{r}))$ with the CLIP region representations $\mathcal{I}(\tilde{r})$,

$$\mathcal{L}_1 = \frac{1}{K} \sum_{\tilde{r}} \| \tilde{\mathcal{R}} - \mathcal{I}(\tilde{r}) \|_1 . \tag{1}$$

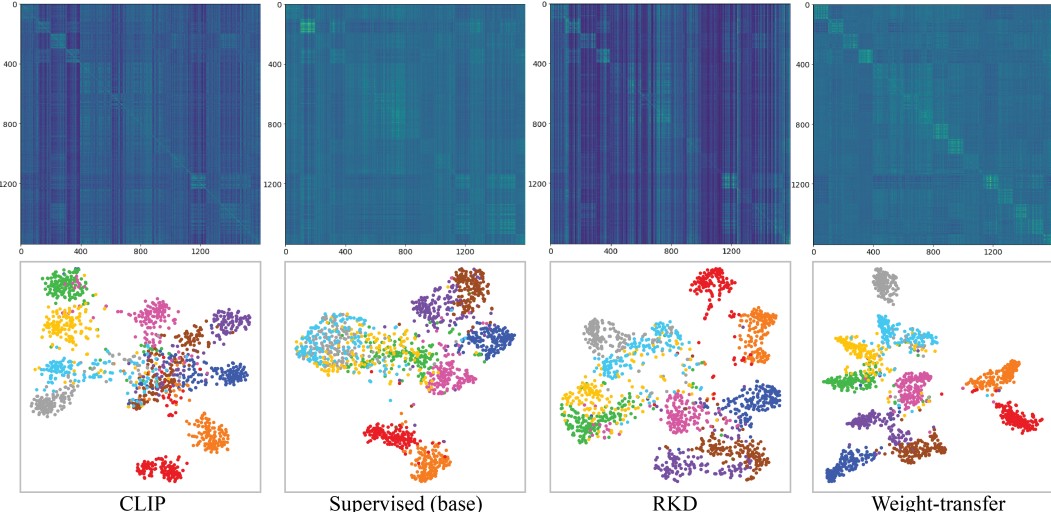

CLIP          Supervised (base)         RKD        Weight-transfer

Figure 2: *Top-row*: Similarity matrices computed on the CLIP ($S_I$) and detector ($S_R$) region embeddings for COCO novel classes. A subset of 100 randomly selected samples per category form a batch represented by a column are grouped together. Our region-based distillation enforces the similarity patterns in the RKD model to be closer to the teacher model, CLIP, indicated by the bright colors along diagonals. *Bottom-row*: t-SNE plots of CLIP and detector region embeddings on novel COCO categories. The CLIP aligned RKD and weight transfer detector embeddings shows improved separability among novel class features as compared to the supervised detector region embeddings (*figure best viewed in-zoom*).

Using this criteria, our visual encoder, along with the VL projection layer $f(\cdot)$, approximates the CLIP image encoder and consequently aligns our region embeddings with the CLIP text embeddings.

**(2)** *Inter-embedding relationship matching loss (IRM):* It is a knowledge distillation based loss $\mathcal{L}_{irm}$ that instills inter-embedding relationships within our region representations to be consistent to the CLIP region representations [39]. Instilling such inter-embedding relations would be beneficial as we know that the teacher model $\mathcal{I}(\cdot)$, and the student model (our detector), are different in nature with respect to their training methods (Fig. 2). The IRM loss is defined on pairwise similarity matrices of the two different sets of embeddings. Specifically, with the top-K proposals computed from $\mathcal{Q}_{\text{pseudo}}$, we compose $K \times K$ similarity matrices for $\mathcal{I}(\tilde{r})$ and $\tilde{\mathcal{R}}$ denoted by $S_I$ and $S_R$ respectively. Notably, these matrices are normalized by L2 norm applied row-wise. The IRM loss is a Frobenius norm $\| \cdot \|_F$, over the mean element-wise squared difference between $S_{\mathcal{I}}$ and $S_R$,

$$S_R = \frac{\tilde{\mathcal{R}} \cdot \tilde{\mathcal{R}}^T}{\| \tilde{\mathcal{R}} \cdot \tilde{\mathcal{R}}^T \|_2}, \quad S_{\mathcal{I}} = \frac{\mathcal{I}(\tilde{r}) \cdot \mathcal{I}(\tilde{r})^T}{\| \mathcal{I}(\tilde{r}) \cdot \mathcal{I}(\tilde{r})^T \|_2},$$

$$\mathcal{L}_{irm} = \frac{1}{K^2} \| S_R - S_{\mathcal{I}} \|_F^2 . \tag{2}$$

We weight the $\mathcal{L}_1$ and $\mathcal{L}_{irm}$ losses by factors $\beta_1$ and $\beta_2$, respectively. Together with the standard two-stage detector losses; RPN loss ($\mathcal{L}_{rpn}$), regression loss ($\mathcal{L}_{reg}$) and classification loss ($\mathcal{L}_{cls}$) [35, 40]; the overall training objective with RKD can be expressed as,

$$\mathcal{L}_{RKD} = \mathcal{L}_{rpn} + \mathcal{L}_{reg} + \mathcal{L}_{cls} + \beta_1 \, \mathcal{L}_1 + \beta_2 \, \mathcal{L}_{irm}. \tag{3}$$

### 3.3   Image-level Supervision with Pseudo Box Labels

In the open-vocabulary setting, a fundamental challenge is to generalize the detector to novel classes. However, due to the daunting task of densely locating all objects in natural scenes, the existing detection datasets are of relatively smaller magnitude compared to the classification datasets, which are easier to annotate. To this end, Zhou *et al.* [2] proposed to take advantage of a large-scale image classification dataset during training to expand the detector's vocabulary. However, an important question is how to effectively associate the region proposals of novel objects with the corresponding labels. We note that the existing approach uses heuristics such as selecting the whole image as a single box, or just the maximum sized box from the RPN, which can ignore potential objects (Fig. 3a).

We propose a weakly-supervised method to generalize the detector to novel categories by using pseudo-box labels from pretrained MViT [8]. We follow [2] to train the detector with a combination

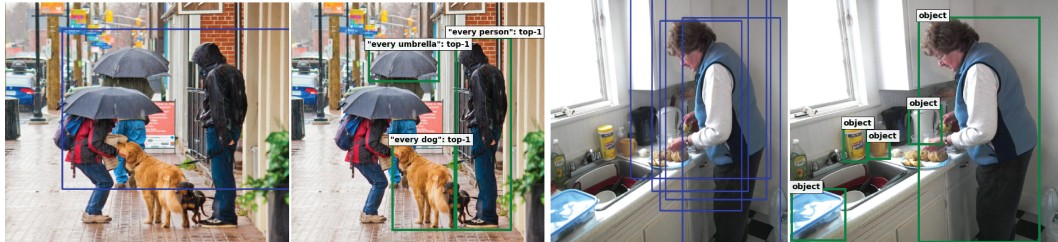

| (a) Class-specific Proposals | (b) Class-agnostic Proposals |
| --- | --- |

Figure 3: **(a) Class-specific Proposals:** A visual comparison of heuristic methods (*left*) used for visual grounding in image-level supervision [2] with our proposed method (*right*). Using heuristic based approaches like selecting maximum sized box from the RPN can ignore local objects in the scene. In our method, we design class-specific text queries with known class labels for pseudo-labeling potential objects. **(b) Class-agnostic Proposals:** In region-based knowledge distillation (RKD), we induce better region-level alignment with fewer high-quality proposals from a generalized class-agnostic proposal generator [8]. We compare top-K RPN proposals (*left*) with top-K multi-modal ViTs proposals used in a class-agnostic manner (*right*).

of detection and classification dataset. A batch of data is prepared by combining data from the detection dataset $\mathcal{D}_{det}$ that are exhaustively annotated with bounding-box and class labels, with data from a classification dataset $\mathcal{D}_{cls}$ that only contains image-level labels. With $\mathcal{Q}_{pseudo}$, we obtain the pseudo-box labels on this classification dataset, which we use for ***image-level supervision*** (ILS). Specifically, consider a sample image $I \in \mathcal{D}_{cls}$, which has a total of $N$ ground-truth class labels, we generate object proposals offline with the use of MViT corresponding to these weak labels. Specifically, we construct $N$ class-specific text queries $\{t_n\}_{n=1}^N$ with template 'every {category}', and obtain $K$ proposals $\{\tilde{r}_k\}_{k=1}^K$ and corresponding confidence scores $\{\tilde{s}_k\}_{k=1}^K$ for each query,

$$[(\tilde{r_1}, \tilde{s_1}), (\tilde{r_2}, \tilde{s_2}), \cdots (\tilde{r_K}, \tilde{s_K})] = \mathcal{Q}_{pseudo}(I, t_n); \;\; I \in \mathcal{D}_{cls}, n \in N.$$

We select the top-1 proposal with the highest confidence score, as the pseudo-box label for a particular category. This gives us $N$ high-quality pseudo-box labels for each image, corresponding to its $N$ image-level category labels (Fig. 3a). We compute the region embeddings $\tilde{\mathcal{R}}$ for proposals $\tilde{r}$ as,

$$\tilde{\mathcal{R}}_n = f(\phi(\tilde{r}_{\hat{k}})), \;\; \hat{k} = \text{argmax}_k(\tilde{s}_k).$$

In the case of $\mathcal{D}_{det}$, the training follows the standard two-stage RCNN training recipe. However, for $\mathcal{D}_{cls}$, only the classification loss is updated. We call this *pseudo-max score*, $\mathcal{L}_{pms}$ loss.

$$\mathcal{L}_{pms} = \frac{1}{N} \sum_n BCE(\mathcal{V}, y_{\tilde{r}}), \text{ where } \mathcal{V} = \cos\left(\tilde{\mathcal{R}}_n, \mathcal{T}\right). \tag{4}$$

We weight $\mathcal{L}_{pms}$ by a factor $\alpha$ and the overall training objective with our ILS can be expressed as,

$$\mathcal{L}_{ILS} = \begin{cases} \mathcal{L}_{rpn} + \mathcal{L}_{reg} + \mathcal{L}_{cls}, & \text{if} \quad I \in \mathcal{D}_{det} \\ \alpha \, \mathcal{L}_{pms}, & \text{if} \quad I \in \mathcal{D}_{cls}. \end{cases} \tag{5}$$

### 3.4   Weight Transfer Function

To combine the alignment from region-based distillation (Sec. 3.2) with the benefits from weak supervision with pseudo-box labels (Sec. 3.3), a naive approach would be to train the detector with a combination of losses: $\mathcal{L}_1$ (1), $\mathcal{L}_{irm}$ (2) and $\mathcal{L}_{pms}$ (4). However, we demonstrate that a simple combination of the two approaches does not lead to complimentary benefits, instead they compete with each other (Table 2). The additional supervision from pseudo-labels improves the generalization of the detector, while the region-based distillation works towards object-centric alignment in the language domain, thereby improving the overall performance of the detector. We aim to incorporate the benefits from the two approaches and preserve the object-centric alignment in the language domain. To this end, we use a weight transfer mechanism [41] from VL projection used in region-based distillation to the weak supervision by learning a ***weight transfer function***, $\mathcal{W}_{\mathcal{T}}(\cdot)$. In other words, the VL projection function $f(\cdot)$ used during the weak image-level supervision is explicitly conditioned on the mapping function used for alignment in the distillation process. This way, both the transformations are tied together to reinforce mutual representation capability and avoid any conflict in the learned function mapping. Let the weights of the projection layer in RKD and weak

image-level supervision be represented as $W_D$ and $W_P$ respectively. The weight transfer operation is given by,

$$W_P = \mathcal{W}_\mathcal{T}(W_D) = \Big(W_{\theta_2}\ \rho(W_{\theta_1}\ W_D)\Big); \qquad \mathcal{W}_\mathcal{T}:\ W_D \rightarrow W_P.$$

Here, $W_D$ is kept frozen and we design $\mathcal{W}_\mathcal{T}$ as a 2-layer MLP, $W_{\theta_1}$ followed by $W_{\theta_2}$ a with LeakyReLU ($\rho$) activation with a negative slope of 0.1. Further, we use a skip connection across $W_P$ by projecting the original representations using a separate 2-layer MLP (Fig. 1). The total loss here is a combination of $\mathcal{L}_{RKD}$ (Eq. 3) and $\mathcal{L}_{ILS}$ (Eq. 5) loss, given by,

$$\mathcal{L} = \mathcal{L}_{rpn} + \mathcal{L}_{reg} + \mathcal{L}_{cls} + \beta_1\ \mathcal{L}_1 + \beta_2\ \mathcal{L}_{irm} + \alpha\ \mathcal{L}_{pms}.$$

# 4  Experiments

## 4.1  Datasets

We conduct our experiments on COCO [42] and LVIS v1.0 [43] under OVD setting. For evaluation, we use the generalized ZSD setting where the classifier contains both base and novel categories. Table 1 summarizes all the datasets used in our work. Following [2, 1], we use a subset of ImageNet-21K having 997 overlapping LVIS categories and COCO captions dataset for ILS in LVIS and COCO experiments respectively (refer

| Dataset | Dataset Type | Task | # images |
|---|---|---|---|
| COCO | Detection | OVD | 118K |
| LVIS v1.0 | Detection | OVD | 100K |
| ImageNet-21K* | Classification | ILS in LVIS | 1.4M |
| COCO-Captions | Image-captioning | ILS in COCO | 118K |
| LMDet | Flickr30, GQA & Visual Genome | MViT Pretraining | 1.1M |
| ‡ LMDet | LMDet (excluding any overlap with novel categories) | MViT Pretraining | 0.8M |

Table 1: Summary of the datasets used in our experiments.

Appendix. A for more details). For the pseudo-labeling process $\mathcal{Q}_{\text{pseudo}}$, we use the MViT pretrained on a Large-scale Modulated Detection (LMDet) dataset [8]. We ensure that MViT pretraining dataset has no overlap with any of the evaluation datasets in our work. Additionally, in all our experiments we use a pretrained MViT that we train using the author's provided code on filtered LMDet (‡LMDet) dataset by entirely restricting any exposure to the novel/rare classes in evaluation.

**COCO OVD:** We use COCO-2017 dataset for training and validation. We follow the ZS splits proposed in [10], in which 48 categories are selected as base and 17 are selected as novel classes.

**LVIS OVD:** LVIS contains 1203 categories which are further split into frequent, common and rare categories. Inline with [6, 2], we combine the frequent and common categories to form base classes and keep all rare classes as novel, resulting in 866 base and 337 rare classes.

**Cross-transfer Datasets:** To validate the adaptability of our method, we evaluate and compare results of our LVIS trained model on OpenImages[44] and Objects365 [45] and COCO [42] datasets.

## 4.2  Implementation details

We conduct COCO experiments using Faster R-CNN [35] with ResNet-50 backbone. We train the supervised-base model on 48 base classes ($\mathcal{C}_B$) for 1x schedule ($\sim$12 COCO epochs) and report box $AP_{50}$. For RKD, we finetune this model for another 1x schedule using box labels from $\mathcal{C}_B$ and class-agnostic proposals from the pretrained MViT [8]. This model is further finetuned for 1x schedule with ILS and the associated weight transfer function using class labels from COCO captions and corresponding class-specific proposals from MViT. This sums to an overall 3x training schedule.

For LVIS experiments, we use Mask R-CNN [40] with federated loss [46] and sigmoid cross-entropy, and report mask AP. For RKD and weight transfer, we use the same training schedules as of COCO and report the average over three runs. For comparison with Detic [2], we apply our proposed method on their strong CenterNetV2 [46] baseline under the same settings. It uses ImangeNet21K pretrained backbone with 4x schedule using large scale jittering (LSJ) [47] augmentations. All of our models are trained using 8 A100 GPUs with an approximate training time of 9 and 6 hours for 1x schedule of COCO and LVIS respectively.

In our experiments, we use SGD optimizer with a weight decay of $1e^{-4}$ and a momentum of 0.9. We train for 1x schedule with batch size of 16 and an initial learning rate of 0.02 which drops by a factor of 10 at the $8^{th}$ and $11^{th}$ epoch. We set temperature $\tau$ to 50. Our longer schedules experiments use 100-1280 LSJ [47]. We use $\alpha$ of 0.1 to weight $\mathcal{L}_{pms}$. For computing CLIP embeddings we use the

CLIP model ViT-B/32 [3], with input size of 224×224. We use the query 'a `photo` of a `{category}`' for to compute the text embeddings for the classifier. For distillation, we use top 5 proposals from the pretrained MViT [8] evaluated with generic query, 'all `objects`', generating class-agnostic proposals. We refer to Appendix D for additional details on the approach we use to generate class-agnostic and class-specific proposals from MViT. In COCO experiments, we set weights $\beta_1$ and $\beta_2$ to 0.15. In LVIS, we set $\beta_1$ to 0.15 and $\beta_2$ to 0.25. We choose these values using a randomized hyper-parameter search on the corresponding held-out datasets. The 2-layer MLP in our weight transfer function has a hidden dim of 512, and a hidden dim of 1024 is used in the MLP skip connection across $W_P$ in Fig. 1 (refer to Appendix C for more details).

## 4.3 Our Approach: Main results

Table 2 shows the contribution of individual components in our proposed approach. Building on top of the supervised-base model, our *region-based knowledge distillation* (RKD) shows an absolute gain of 19.5 and 1.5 AP for COCO novel and base classes respectively, indicating the adaptability of image-centric CLIP embeddings for local regions. With *pseudo-box labeled weak image-level supervision* (PIS), novel class AP improves by 28.7, demonstrating generalization to novel classes and thus enlarging the detector's vocabulary. Naively combining the two approaches shows improvement, but struggles to maintain the gains from the individual components. In contrast, our *weight transfer* method suitably combines the complimentary benefits of both components (Fig. 2), achieving 36.6 AP on novel classes while maintaining performance on base classes.

| Method | $AP_{novel}$ | $AP_{base}$ | AP |
|---|---|---|---|
| 1: Supervised (Base) | 1.7 | 53.2 | 39.6 |
| 2: Base + Region based ditillation (RKD) | 21.2 | **54.7** | 45.9 |
| 3: Base + ILS with pseudo-box (PIS) | 30.4 | 52.6 | 46.8 |
| 4: RKD + PIS | 31.5 | 52.8 | 47.2 |
| 5: RKD + PIS + Weight-transfer (Ours) | **36.6** | 54.0 | **49.4** |

Table 2: Effect of individual components in our method. Our weight transfer method provides complimentary gains from RKD and ILS, achieving superior results as compared to naively adding both components.

**Open-vocabulary Detection - COCO:** We compare our OVD results with previously established methods in Table 3. OVR-CNN learns a vision-to-language mapping with expensive pretraining. Detic uses ILS to improve detection on novel classes. We use a novel weight transfer function to perform object-centric VL alignment and achieve 54.0 AP on the base classes, surpassing OVR-CNN and Detic by 8.0 AP and 0.2 AP respectively. On novel classes our method achieves 36.6 AP, the highest novel AP achieved over all methods. In comparison with ViLD, which trains for 8x schedule ($\sim$ 96 epochs), our method with the same schedule provides 56.6 base AP, lagging by 2.9.

| Method | Supervision | $AP_{base}$ | $AP_{novel}$ | AP |
|---|---|---|---|---|
| WSDDN§ [24] | image-level labels for $\mathcal{C}_B \cup \mathcal{C}_N$ | 19.6 | 19.7 | 19.6 |
| Cap2Det§ [27] | | 20.1 | 20.3 | 20.1 |
| OVR-CNN [1] | pretraining with captions $\mathcal{C}_B \cup \mathcal{C}_N$ box-level labels in $\mathcal{C}_B$ | 46.0 | 22.8 | 39.9 |
| ViLD† [6] | internet sourced image-text pairs box-level labels in $\mathcal{C}_B$ | **59.5** | 27.6 | 51.3 |
| RegionCLIP [7] | internet sourced image-text pairs pretraining with pseudo box-level labels box-level labels in $\mathcal{C}_B$ | 54.8 | 26.8 | 47.5 |
| Detic [2] | internet sourced image-text pairs image-level labels for $\mathcal{C}_B \cup \mathcal{C}_N$ box-level labels in $\mathcal{C}_B$ | 47.1 | 27.8 | 45.0 |
| Detic‡ | | 53.8 | 28.4 | 47.2 |
| Ours | internet sourced image-text pairs image-level labels for $\mathcal{C}_B \cup \mathcal{C}_N$ pseudo-box labels in $\mathcal{C}_N$, box-level labels in $\mathcal{C}_B$ | **54.0** | **36.6** | **49.4** |
| Ours † | | 56.6 | 36.9 | 51.5 |

Table 3: **OVD results on COCO.** Here $\mathcal{C}_B$ and $\mathcal{C}_N$ represents the base and novel classes respectively. §The results quoted from [1]. †ViLD and our methods are trained for longer 8x schedule (shown in gray). ‡We train detic for another 1x for a fair comparison with our method. For ViLD, we use their unified model that trains ViLD-text and ViLD-Image together. For Detic, we report their best model.

On novel classes, we achieve 36.9 AP surpassing ViLD by a gain of 9.3. In contrast to ViLD design, our weight transfer function allows both RKD and ILS to provide complimentary gains without any negative competition among the two methods [6].

**Open-vocabulary Detection - LVIS:** Table 4 (left) compares our results with ViLD [6] on LVIS benchmark. With 3x training schedule ($\sim$ 36 epochs) we perform reasonably well compared to ViLD 32x schedule ($\sim$ 384 epochs), already surpassing the rare AP by 1.0 while having slightly lower performance on frequent classes. Extending our model to 8x schedule fills the gap, surpassing ViLD by 0.8 in frequent and 5.0 AP in rare classes respectively. In Table 4 (right), we compare our method with Detic by using their strong LVIS baseline that uses CenterNetV2 network. Following similar settings, we finetune their box-supervised model using our weight transfer method and show improvements.

| Method | Epochs | $AP_r$ | $AP_c$ | $AP_f$ | AP |
|--------|--------|--------|--------|--------|-----|
| ViLD [6] | 384 | 16.1 | 20.0 | 28.3 | 22.5 |
| Ours | 36 | 17.1 | 21.4 | 26.7 | 22.8 |
| Ours | 96 | **21.1** | **25.0** | **29.1** | **25.9** |

| Method | $AP_r$ | $AP_c$ | $AP_f$ | AP |
|--------|--------|--------|--------|-----|
| Box-supervised [2] | 16.3 | 31.0 | 35.4 | 30.0 |
| Detic (Image + Captions) | 24.6 | 32.5 | 35.6 | 32.4 |
| Ours | **25.2** | **33.4** | **35.8** | **32.9** |

Table 4: **OVD results on LVIS.** (*Left*): Comparison with prior work ViLD, using their unified model (ViLD-text + ViLD-Image), show improvement across novel and base categories. (*Right*): We show the comparison with Detic, by building on their strong LVIS baseline using CenterNetV2 detector [2]

**Strict Open-vocabulary Setting:** Inspired from Detic, we define our work under the weakly-supervised open-vocabulary setting as it uses image-level labels for expanding the detector's vocabulary. However in this setting, the complete target vocabulary set is unknown, *i.e.*, only a selected number of novel and base categories are used for ILS from ImageNet-21K in LVIS. To evaluate our model in an extensive open-vocabulary setting, we modify our

| Method | Epochs | $AP_r$ | $AP_c$ | $AP_f$ | AP |
|--------|--------|--------|--------|--------|-----|
| ViLD [6] | 384 | **16.1** | 20.0 | **28.3** | **22.5** |
| Ours | 36 | 16.0 | **20.2** | 26.3 | 21.8 |

Table 5: Performance on LVIS benchmark using a strict OVD setting.

ILS by considering a larger vocabulary. Specifically, we expand the vocabulary to five times its size in [2], by applying ILS from randomly sampled 5K categories from ImageNet-21k, in addition to the LVIS base classes. Table 5 compares our strict OVD setting results with ViLD where our performance slightly degrades showing sensitivity to ILS. However, we expect a gain with longer training as in Table 4. In addition to above two settings, we train our LVIS model under stricter OVD conditions in a *non* weakly-supervised setting by only using LVIS base categories for ILS. We achieve an overall 21.71 AP which is close to the model trained using ILS from 997 categories (22.75 AP).

**Cross-dataset evaluation performance:** We provide cross-dataset evaluation of our model in Table 6 and compare with prior OVD works. ViLD-text[6] and Detic-base[2] are box-supervised baseline models for ViLD and Detic respectively. Our method builds on top of Detic-base and shows favourable results when directly transferred to cross-datasets without any dataset-specific finetuning. We use our method trained on LVIS and report $AP_{50}$ on COCO [42], OpenImages [44] and Objects365 [45].

| Method | COCO | OpenImages | Objects365 |
|--------|------|------------|------------|
| ViLD-text | 43.4 | - | 11.1 |
| Detic-base† | 55.3 | 37.4 | 19.2 |
| ViLD | 55.6 | - | 18.2 |
| Detic† | 56.3 | 42.2 | 21.7 |
| Ours | **56.6** | **42.9** | **22.3** |

Table 6: Cross-dataset evaluation. †The results evaluated using official implementation.

### 4.4 Analysis of RKD and ILS

**Effect of Region-based Knowledge Distillation (RKD):** We ablate the effect of $\mathcal{L}_1$ (Eq. 1) and $\mathcal{L}_{irm}$ (Eq. 2) RKD approach on COCO (Table 7). The results show the importance of both loss functions, where using $\mathcal{L}_1$ loss over base model with top-5 proposals from MViT [8] improves the base and novel class by 1.9 and 15.0 AP (row-1 vs 3). Using $\mathcal{L}_{irm}$ in row-4 further improves the overall and novel class AP. To show the importance of using quality proposals in RKD, we compare the model trained with $\mathcal{L}_1$ loss using top-5 RPN vs MViT proposals (row-2 vs 3). All the models in rows 2-4 are finetuned on the base model.

**Effect of Weak Image-level Supervision (ILS):** We compare different choices of ILS in Table 8. Our $\mathcal{L}_{pms}$ loss (Eq. 4) is compared with previously adopted ILS approaches [31, 32, 2] (rows 2-3). In

| Method | AP_novel | AP_base | AP |
|---|---|---|---|
| 1: Supervised (Base) | 1.7 | 53.2 | 39.6 |
| 2: RPN proposals $\mathcal{L}_1$ loss | 4.0 | 54.9 | 41.6 |
| 3: MViT prop - $\mathcal{L}_1$ loss | 16.7 | **55.1** | 45.0 |
| 4: $\mathcal{L}_1$ + IRM loss | **21.2** | 54.7 | **45.9** |

Table 7: Analysis on our region-based KD.

| Method | AP_novel | AP_base | AP |
|---|---|---|---|
| 1: Supervised (Base) | 1.7 | **53.2** | 39.6 |
| 2: Max-Score loss on RPN | 15.9 | 48.2 | 39.7 |
| 3: Max-Size loss on RPN | 25.9 | 51.1 | 44.5 |
| 4: Max-Size of MViT | 28.9 | 50.7 | 45.0 |
| 5: Pseudo-box on MViT | **30.4** | 52.6 | **46.8** |

Table 8: Analysis on our weak IL supervision.

row-4, we generate class-agnostic object proposals using 'all objects' text query with multi-modal ViTs (MViTs) [8] and select max-size proposal for ILS. In row-5, our proposed ILS approach uses target specific 'every {category}' text query with MViT and selects top-1 proposal for each ILS category. Our method (row-5) shows better performance compared to other alternatives. Additionally, we present all ablations on LVIS dataset in Appendix C.

## 5 Qualitative Results

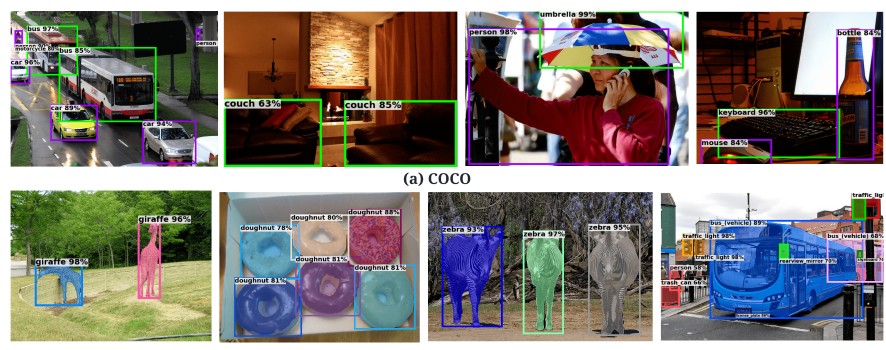

Figure 4: Qualitative results on (**a**) COCO and (**b**) LVIS images. For COCO, base and novel categories are shown in purple and green colors respectively.

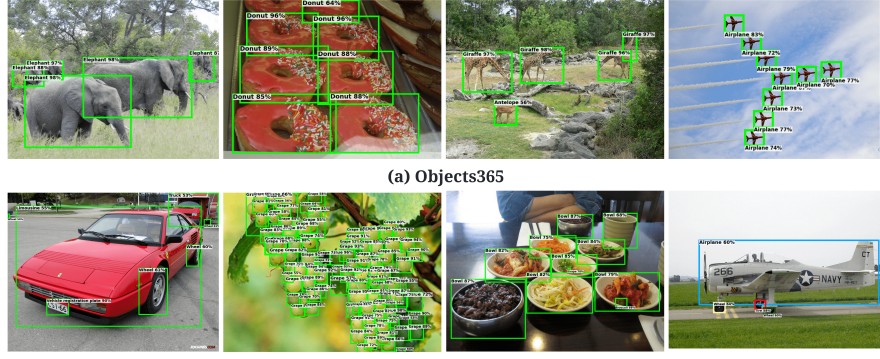

Figure 5: Qualitative results of cross-dataset transfer of our LVIS OVD model on (**a**) Objects365 and (**b**) OpenImages. Without any finetuning, our method provides high-quality detections.

## 6 Conclusion

This paper develops a novel framework to leverage the representation and generalization capability of pre-trained multi-modal models towards improved open-vocabulary detection (OVD). Specifically, we note that the existing OVD methods use weak supervision modes that are more image-centric, rather than object-centric for the end detection task. We proposed a novel knowledge distillation approach together with object-level pseudo-labeling to promote region-wise alignment between visual and language representations. Our weight transfer module provide an integration mechanism to combine the benefits of knowledge distillation and object-level pseudo-labeling. We demonstrate encouraging results on four popular OVD benchmarks, demonstrating sound generalization ability.

**Acknowledgements:** The computations were performed in the Berzelius resource provided by the Knut and Alice Wallenberg Foundation at the National Supercomputer Centre.

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
