# Supplemental Material

In this section, we provide additional information regarding,

- Implementation details (Appendix A)
- Qualitative Results (Appendix 5)
- Zero-shot Region Classification (Appendix B)
- Additional Ablation Experiments (Appendix C)
- Pseudo-labeling using Multi-modal ViTs (Appendix D)
- Limitations (Appendix E)
- Potential Negative Social Impacts (Appendix F)
- Ethical Considerations (Appendix G)
- Datasets License Details (Appendix H)

## A   Implementation Details

We provide additional implementation details for our approach and datasets used in this work. We use standard Faster R-CNN [35] with ResNet-50 C4 backbone and Mask R-CNN [40] with ResNet-50 FPN backbone for COCO and LVIS experiments respectively. We use L2 normalization on the region and text embeddings before computing the RKD loss and final classification scores. We note that this normalization is helpful to stabilize the training. For ILS, we sample images from detection and classification datasets with a ratio of 1:4. Specifically, we use a batch size of 16 and 64 for detection and classification datasets, respectively. We will release our codes and pretrained models publicly to ensure reproducibility of our results.

**Datasets for weak Image-level Supervision (ILS):** We use COCO captions and ImageNet-21k [4] datasets for our proposed Image Level supervision (ILS) on COCO and LVIS datasets respectively. COCO captions dataset uses images from COCO detection dataset and provides five captions for each image. The words in a caption are compared heuristically, with every category name in the list of categories in COCO (base + novel). Using this method, we generate a list of positive categories for each image which is used as labels for ILS. We use ImageNet-21k [48] for LVIS experiments which is a large scale classification dataset containing approximately 14M images and 21K classes. We use categories from ImageNet-21k which overlaps with LVIS categories, resulting in a subset containing 997 categories.

**Cross-dataset evaluation:** We provide cross-dataset evaluation of our LVIS trained model in Table 6. Following [2, 6], we use validation sets of OpenImages V5 containing ∼41K images and Objects365 V2 containing ∼80K images for evaluation. We report $AP_{50}$ for cross-data evaluation.

## B   Zero-shot Region Classification

We compare the zero-shot classification performance of open-vocabulary detector with pretrained CLIP [3] model on COCO validation dataset. Table 9 shows the results where the top-1 classification accuracy is evaluated using the ground-truth object bounding boxes from COCO. The CLIP pretrained model shows better results for novel classes as compared to supervised-base model, indicating the strong generalization of the CLIP (row-1 vs 2). However the base class accuracy is higher for the supervised-base model as it is trained using COCO base classes. Further, using our region-based knowledge distillation (RKD) and novel weight transfer function improves the base and novel class performance, indicating the object-centric alignment in latent space.

## C   Additional Ablation Experiments

### C.1   Ablation Experiments on LVIS

**Effect of individual components:** Table 10 shows the contribution of individual components in our proposed approach on LVIS dataset. The baseline Mask-RCNN model (row-1) is trained on LVIS frequent and common classes using only the box-level supervision along with the zero-shot CLIP [3] classifier. The results indicate the effectiveness of our region-based distillation (RKD)

| Method | Top-1$_{base}$ | Top-1$_{novel}$ | Top-1$_{overall}$ |
|---|---|---|---|
| 1: Supervised (Base) | 88.8 | 42.5 | 76.7 |
| 2: CLIP | 57.3 | 59.4 | 57.8 |
| 3: RKD | 86.0 | 60.2 | 79.2 |
| 4: Weight transfer | 90.3 | 82.2 | 88.2 |

Table 9: Classification results on novel and base classes with boxes cropped from COCO validation dataset using ground truth annotations. The pretrained CLIP shows competitive novel class accuracy. Our proposed RKD and weight transfer approach further improve the performance.

which explicitly aligns the image-centric CLIP embeddings to the object-centric region embeddings. Our image-level supervision (ILS) which uses class-specific pseudo-labels from the pretrained multi-modal ViT [8], effectively enlarges the detector's vocabulary indicated by an increase of 4.8 AP over the base model for rare categories. Further, our proposed weight transfer scheme combines the strengths of the two methods and achieves better results on the common and frequent categories, while performing on par for the rare classes compared to naively combining the two approaches (row-4 vs 5).

| Method | AP$_r$ | AP$_c$ | AP$_f$ | AP |
|---|---|---|---|---|
| 1: Supervised (Base) | 12.2 | 19.4 | 26.4 | 20.9 |
| 2: Base + Region based ditillation (RKD) | 15.2 | 20.2 | 27.3 | 22.1 |
| 3: Base + ILS with pseudo-box (PIS) | 17.0 | 21.2 | 26.1 | 22.4 |
| 4: RKD + PIS | 17.3 | 20.9 | 25.5 | 22.1 |
| 5: RKD + PIS + Weight-transfer (Ours) | 17.1 | 21.4 | 26.7 | 22.8 |

Table 10: Effect of individual components in our method on LVIS dataset. Using RKD provides improvements over the baseline in all metrics (row-1 vs 2). Using ILS mainly helps in improving rare class performance (row-1 vs 3). Simply combining two methods shows improvements over the baseline but struggles to retain the individual performances especially for common and frequent categories (row-4). Our weight transfer approach provides complimentary gains from RKD and ILS, achieving good results as compared to simply adding both components (row-4 vs 5).

**Effect of Region-based Knowledge Distillation (RKD):** Table 11 shows the effect of different loss functions ($\mathcal{L}_1$ and $\mathcal{L}_{irm}$ in Eq. 1 and Eq. 2 respectively) used in our region-based knowledge distillation (RKD) on LVIS dataset. It shows the effectiveness of using proposals from multi-modal ViT (MViT) [8] as compared to RPN for region-level alignment (row-2 vs 3). Using high-quality MViT proposals provides significant gains compared to using RPN proposals. Further, using our inter-embedding relationship matching (IRM) loss along with $\mathcal{L}_1$ loss provides an overall good trade-off between rare, common and frequent class AP.

| Method | AP$_r$ | AP$_c$ | AP$_f$ | AP |
|---|---|---|---|---|
| 1: Supervised (Base) | 12.2 | 19.4 | 26.4 | 20.9 |
| 2: RPN proposals $\mathcal{L}_1$ loss | 8.7 | 17.4 | 26.1 | 19.3 |
| 3: MViT prop - $\mathcal{L}_1$ loss | 12.4 | 20.7 | 27.7 | 22.0 |
| 4: $\mathcal{L}_1$ + IRM loss | 15.2 | 20.2 | 27.3 | 22.1 |

Table 11: Analysis on our RKD method on LVIS.

**Effect of Weak Image-level Supervision (ILS):** Table 12 compares the different heuristics based approaches opted for image-level supervision (ILS) versus our method that utilizes class-specific proposals from the pretrained MViT on LVIS dataset. Selecting top-1 proposal from MViT using target specific specific queries such as 'every {category}' provides optimal performance for rare classes.

| Method | $AP_r$ | $AP_c$ | $AP_f$ | AP |
|---|---|---|---|---|
| 1: Supervised (Base) | 12.2 | 19.4 | 26.4 | 20.9 |
| 2: Max-Score loss on RPN | 12.8 | 18.6 | 24.7 | 20.0 |
| 3: Max-Size loss on RPN | 14.9 | 21.3 | 26.1 | 22.1 |
| 4: Pseudo-box on MViT | 17.0 | 21.2 | 26.1 | 22.4 |

Table 12: Analysis on our weak ILS on LVIS.

## C.2 Initialization for RKD Training

We note that it is important to properly initialize the RKD training to gain its full advantages. Table 13 shows that training RKD from scratch (row-2) results in lower base class AP. However, initializing the RKD training from the Supervised base model recovers this loss and provides improvements over the base model. This indicates that region-based alignment is sensitive to the distribution of the features and requires mature features for effectively distilling knowledge from pretrained CLIP model. This observation is same as in [49] where the contrastive clustering is enabled only on the mature features after a few training epochs for open-world object detection.

| Method | $AP_{novel}$ | $AP_{base}$ | AP |
|---|---|---|---|
| 1: Supervised (Base) | 1.7 | 53.2 | 39.6 |
| 2: RKD from scratch | 21.3 | 50.9 | 43.1 |
| 3: Base + RKD | 21.2 | 54.7 | 45.9 |

Table 13: Effect of initialization for RKD training on COCO dataset.

## C.3 Additional Ablation Experiment

Table 14 shows the ablation on using a MLP skip connection across $\mathcal{W}_{\mathcal{P}}$ in Fig. 1. We add this skip connection to form a direct path for region classification using CLIP in ILS. This allows the weight transfer function to specifically focus on the residual signal in the ILS pathway. It improves the convergence and helps to attain better results in most cases on LVIS/COCO datasets.

| | COCO | | | LVIS | | | |
|---|---|---|---|---|---|---|---|
| Method | $AP_{novel}$ | $AP_{base}$ | AP | $AP_r$ | $AP_c$ | $AP_f$ | AP |
| 1: Supervised (Base) | 1.7 | 53.2 | 39.6 | 12.2 | 19.4 | 26.4 | 20.9 |
| 2: RKD + PIS + Weight-transfer (Ours) | 36.6 | 54.0 | 49.4 | 17.1 | 21.4 | 26.7 | 22.8 |
| 3: + w/o MLP skip connection | 32.5 | 53.5 | 48.0 | 18.1 | 20.9 | 26.2 | 22.5 |

Table 14: The ablation on using MLP skip connection in Fig. 1.

## D Pseudo Labeling using Multi-modal ViTs

In this section, we describe the process of generating class-agnostic and class-specific proposals using multi-modal ViTs (MViTs) [8, 50]. We name this process as *pseudo labeling* $\mathcal{Q}_{pseudo}$. The MViT model is trained using aligned image text pairs and is capable of locating novel and base class objects using relevant human-intuitive text queries. For example, targeted text queries such as 'every person' and 'every elephant' can be used to locate all persons and all elephants in an image respectively (Fig. 6b). Maaz *et al.* [8] show that the MViTs encode the object-centric concepts using aligned image-caption pairs and are excellent class-agnostic object detectors. The authors designed text queries such as 'all objects' and 'all entities' and demonstrated state-of-the-art class-agnostic object detection results on multiple datasets across different domains. We use these MViTs to generate class-agnostic and class-specific object proposals for region-based knowledge distillation (RKD) and weak image-level supervision (ILS), respectively.

**Class-agnostic proposals for RKD:** We generate class-agnostic object proposals from the MViT [8] using 'all objects' text query. The generated proposals are ranked using predicted objectness scores

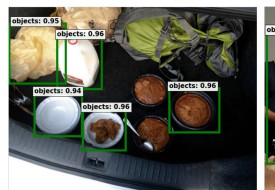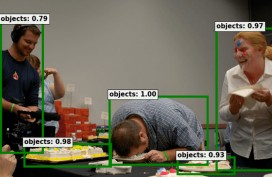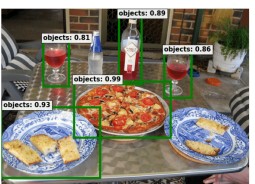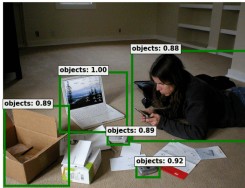

(a) Class-agnostic Proposals (RKD)

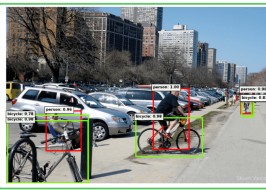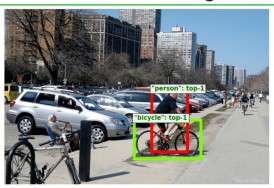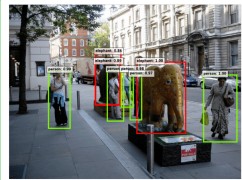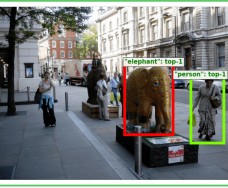

(b) Class-specific Proposals (ILS)

Figure 6: **(a) Class-agnostic Proposals:** The figure shows the top 5 class-agnostic proposals obtained from the MViT [8] using 'all objects' text query. As illustrated, these high-quality tightly bound object proposals provide rich local-semantics for RKD in our proposed pipeline. **(b) Class-specific Proposals:** The figure shows the class-specific proposals obtained from the MViT using 'every <category name>' text queries. The left image in each pair shows all proposals while the corresponding right image shows the selected top 1 proposal per category for ILS.

and the top 5 proposals per image are selected for RKD as shown in Fig. 6a. Next, the CLIP [3] image-encoder and our OVD detector is used to generate embeddings corresponding to these proposals which are then used for calculating the RKD loss in Eq. 3. To save the computation load and increase the training efficiency, we compute the class-agnostic proposals and the corresponding CLIP region embeddings offline and load them during training. Further for LVIS experiments, we use images from a subset of ImageNet-21K (consisting of 997 overlapping LVIS categories) for RKD as well.

**Class-specific proposals for ILS:** We generate class-specific proposals from the MViT [8] using 'every <category name>' text query. Given the $N$ category names present in an image, we use $N$ queries of format 'every <category name>' to generate class-specific proposals followed by selecting top 1 proposal for each category. This provides us $N$ high-quality box proposals per image corresponding to $N$ categories present in the image. These proposals are used to effectively enhance the detector's vocabulary using ILS during training. Further, to maintain the training efficiency of our experiments, we compute these class-specific proposals offline and load them during training.

# E   Limitations

Our proposed OVD method encourages object centric visual-language (VL) alignment using a novel weight transfer method which combines benefits from RKD and ILS. Irrespective of the state-of-the-art results on novel/rare classes, there is still a significant gap between base and novel class performances (e.g. 56.7 and 40.5 AP for COCO base and novel categories in Table 3, 29.1 and 21.1 Mask AP for LVIS frequent and rare categories in Table 4). Further, the open-vocabulary capabilities of our model largely depends or are limited to the vocabulary of the pretrained CLIP [3] model, which is used as a teacher in our RKD pipeline.

# F   Potential Negative Social Impacts

The results of cross-dataset transfer evaluations show that the vocabulary of our detector is highly flexible and can be expanded to any number of categories, based on the downstream tasks and datasets. This poses a risk on how our OVD detector with a large vocabulary can be used in inappropriate ways in the community such as for large scale illegal video surveillance. Furthermore, OVD capabilities can be modulated for targeted detections instead of generic detections by tuning the classifier weights using specialized prompts. This could add biases in the detector and can lead to unfair predictions.

# G   Ethical Considerations

The OVD response to recognize object categories strongly depends on the image-text pretraining datasets used for the training of VL model (CLIP in our case). Thus, the source of these datasets can

pose ethical issues. For example, datasets extracted from internet can contain racial and unethical bias and can modulate the ethical behaviour of the detector as well. Thus, before applying our OVD detector in a practical scenario, such biases of the pretraining/training datasets should be removed to have fairness and ethically correct results of the detector. Moreover, the detector vocabulary is flexible and it can be tuned to show racial biasness while detecting humans. For example, weights of the zero-shot classifier generated with specialized biased prompts could lead to biased and unethically targeted human detections (e.g., black vs white) which must be taken into consideration.

## H   License Details

Here we provide license details of the datasets used in our work, summarized in Table 15. COCO is available for non-commercial use under the Creative Commons Attribution 4.0 (CC BY 4.0) license. LVIS is based on the COCO dataset, and it is licensed under both CC BY 4.0 and the COCO license. ImageNet-21k is a publically available dataset available for research and non-commercial use. It is licensed under Creative Commons (CC), and its type is "CC BY-NC". We use a pretrained MViT model for proposal generation, which is trained on LMDet (Large scale Modulated Detection dataset). It uses Flicker30k, Visual Genome, and GQA datasets. The license type of Flicker30k is CC BY-NC. Visual Genome and GQA both have the same license type CC BY 4.0. For cross-datasets evaluation, Objects365 and OpenImages are used, which are licensed under Creative Commons Attribution 4.0. Annotations of OpenImages are licensed by Google LLC under Creative Commons Attribution 2.0.

| Dataset | Task | License |
|---|---|---|
| COCO | OVD | Custom (CC BY 4.0) |
| LVIS v1.0 | OVD | CC BY 4.0 & COCO license |
| ImageNet-21K | ILS in LVIS | CC BY-NC |
| Flickr30k | MViT | CC BY-NC |
| Visual Genome | MViT | CC BY 4.0 |
| GQA | MViT | CC BY 4.0 |
| Objects365 | Cross-data evaluation | CC BY 4.0 |
| OpenImages | Cross-data evaluation | CC BY 4.0 |
| OpenImages annotations | Cross-data evaluation | Google LLC & CC BY 2.0 |

Table 15: Summary of licenses for datasets used in our experiments.