# OpenReview forum: "Bridging the Gap between Object and Image-level Representations for Open-Vocabulary Detection"
_NeurIPS.cc/2022/Conference — NeurIPS 2022 Accept_

### Official Review · Reviewer_dzUw · 2022-06-24

**Rating:** 4
**Confidence:** 5
**Soundness:** 3 good
**Presentation:** 3 good
**Contribution:** 3 good

**Summary:**

This work aims at training open-vocabulary object detectors by leveraging CLIP and pretrained multi-modal ViT. This work propose RKD to align the region representations from the RPN and that of CLIP image encoder. In order to better make use of external images (like ImageNet) which only provide image-level annotations, this work proposes to use MViT to generate pseudo boxes and labels and use them to self-train the detector. In addition, this work proposes to adopt a weight transfer function to alleviate the competition between the two losses. The results on COCO has a significant improvement over previous works.

**Questions:**

1. What is the performance on MViT on COCO or LVIS in the open-vocabulary setting?
2. Do the author have some results without the use of MViT? I would like to see a fair comparison to Detic.

**Limitations:**

Yes.

**Strengths And Weaknesses:**

Pros
1. The technical contents of the paper are quite a lot. I can tell that the authors put many efforts on the paper. It includes improvements on both knowledge distillation side (as in ViLD) and pseudo boxes labeling side (as in Detic). It is the first paper in attempt to combine the advantages of both knowledge distillation and self-training.
2. The experiments are throughout, including results on different dataset and the ablation study.
3. The use of weight transfer in this task is interesting and shown to help.

Cons
1. My major concern is the use of MViT that makes the comparison against ViLD and Detic unfair. Because (1) model perspective: MViT can already detect objects, so using MViT in this task is like supervising a detector using another trained detector; (2) data perspective: MViT is pretrained on LMDet dataset, which is very large with human annotations. If the authors can address this concern of mine, I may consider raising the rating.
2. The overall pipeline is quite complex, including the three training stages and how different losses/components work together. Moreover, the hyper-parameters need to be searched for different datasets, which makes me concern about the generalization of the proposed method.
3. It feels like the authors combines many existing methods/models in order to achieve good performance. For example, the IRM loss is just the loss proposed in Similarity-Preserving Knowledge Distillation [39]. Without IRM, the proposed RKD degrades to the knowledge distillation proposed in ViLD. Besides, the weight transfer function is proposed in [41].
4. The total training on COCO takes 36 epochs, which is longer than Detic.
5. The advantage of the proposed method on large scale datasets seems insignificant. Although the performance gain on COCO is significant, but the gap between the proposed method and Detic on LVIS and Object365 is marginal (ie. less than 1AP).
6. There are too many abbreviations in the paper, such as RKD, ILS, PIS, IRM loss, which to some extent makes the paper less easy to read.

---

> ### Author Response · Authors · 2022-08-02
> **Response to dzUw**
>
> ## Use of MViT in our pipeline
> The previous works inline with weakly-supervised and OVD uses off the shelf pretrained models, to enhance the performance. For example, WSDNN uses proposals from selective search and edge boxes. Similarly Detic and ViLD use a pretrained Faster-RCNN to generate proposals. In a similar way, we enhance our proposal quality by incorporating MViT in our pipeline without any significant overhead. The approach of using detectors to produce quality proposals is consistent in literature.
>
> Though the MViT training dataset has no overlap with the evaluation data, some novel classes are exposed during MViT training. To this end, we train MViT again using the author's provided code by entirely restricting any exposure to the novel classes. The results in Tables A-2 and A-3 show that even with the exclusion of novel classes in MViT training, our approach consistently maintains a significant advantage over Detic and ViLD on COCO and LVIS datasets.
>
> |Method|AP$_r$|AP$_c$|AP$_f$|AP|
> |-|-|-|-|-|
> |ViLD (384 epochs)|16.1|20.0|**28.3**|22.5|
> |Ours (36 epochs)|**17.1**|**21.4**|26.7|**22.8**|
> Table A-2: Our LVIS results after excluding rare classes from MViT training
>
> |Method|AP$_{novel}$|AP$_{base}$|AP|
> |-|-|-|-|
> |Detic|28.4| 53.8| 47.2|
> |Ours|**36.6**|**54.0**|**49.4**|
> Table A-3: Our COCO results after excluding novel classes from MViT training
> ##  MViT direct evaluation in OVD setting
> We evaluate the performance of MViT directly on the OVD setting. From Table A-4, we observe that naively using and evaluating MViT for OVD task results in lower performance (row-2). However class-agnostic are relatively higher (row-3) suggesting that the classification performance of MViT is originally very low.
>
> |Method|AP$_{novel}$|AP$_{base}$|AP|
> |-|-|-|-|
> |Supervised (Base)|1.7|53.2|39.6|
> |MViT (Class-specific)|5.6|4.1|4.5|
> |MViT (Class-agnostic)|18.5|28.0|25.5|
> |Ours|**40.3**|**54.1**|50.5|
> Table A4: Evaluating MViT directly on OVD COCO
>
> MViT class-specific and class-agnostic proposals can provide better results over supervised base model for novel classes. However it significantly lags over our final results as our other contributions like RKD (Region-based Knowledge Distillation), PIS (Pseudo-labeling for Image-level Supervision) and Weight-transfer ensure the synergy between complementary components, leading to best performance.
> ## Overall pipeline is complex
> Although the method is complex, the proposed multi-phase training pipeline is effective to ensure the synergies between different proposed methods including RKD, PIS and weight transfer. An important contribution of our approach is a careful design for unifying the advantages of generic object detector (MViT), knowledge distillation and pseudo-labeling. It performs well on the cross-dataset evaluation (Table 5 main paper) which indicates the generalization of our model. Our code and pretrained models will be publicly released.
> ## Combining different approaches from literature
> We are the first to simultaneously integrate object-centric visual and language alignment within a single architecture for OVD. This combination is not trivial and combining both techniques naively results in inferior performance as the RKD and PIS objectives compete. Our approach of using RKD along with PIS using weight transfer function mitigates competition and provides complementary benefits. Overall, our proposed approach requires meticulous design to ensure synergy between complementary components existing in the literature leading to good performance improvements.
> ## Training epochs
> For a fair comparison with Detic, we train Detic for 3x schedule (36 epochs).
> ## Small improvements
> Our approach consistently improves over SoTA for all datasets. In comparison to COCO, the gains on LVIS and Objects365 may seem relatively less, however it should be noted that these large-scale datasets are challenging and even small improvements here demonstrate the scalability of our approach.
>
> ## Too many abbreviations
> It will be clarified in the final version.
>
> ## Comparison with Detic (without MViT)
>
> For stricter comparison with Detic, we train our models using RPN proposals instead of MViT proposals (Table A-4). In row-2, we use RPN proposals for RKD. For ILS, we compare max-size proposal (same as Detic) on RPN (row-3) with MViT proposals (row-4). With $\mathcal{Q}_{\text{pseudo}}$, the multi-modal nature of MViTs are be exploited for pseudo-labelling using intuitive text queries (row-5). Finally we combine our proposed RKD and weight-transfer on RPN proposals (row-6) instead of MViT proposals plus Q_pseudo. These elements still show improvement over performance of Detic.
>
> |Method|AP$_{novel}$|AP$_{base}$|AP|
> |-|-|-|-|
> |Detic (3x schedule)|28.4|53.8|47.2|
> |RKD on RPN|12.3|53.9|43.1|
> |Max-size ILS on RPN|25.9| 51.1|44.5|
> |Max-size ILS on MViT|28.9| 50.7|45.4|
> |RKD+ILS+Weight transfer on RPN|31.1|54.0|48.0|
> Table A-4: Comparison with Detic (without MViT) using only RPN

---

### Official Review · Reviewer_wmjL · 2022-07-05

**Rating:** 5
**Confidence:** 5
**Soundness:** 2 fair
**Presentation:** 4 excellent
**Contribution:** 3 good

**Summary:**

This paper proposes an approach for training open-vocabulary object detection models, i.e. models that can detect objects based on a list of object categories that is given at test time, potentially including novel categories not seen during training.

The paper starts from what has become the standard open-vocabulary approach: A Faster-RCNN detection architecture that is combined with a CLIP image/text model to allow open-vocabulary classification. The paper points out that the CLIP model is trained using only image-level supervision, and proposes a two-stage training approach to adapt it for the object-level detection task:

**First training stage: Region-based knowledge distillation:** In this stage, a pre-trained object detector is used to predict class-agnostic boxes on some dataset. A vision/language model (CLIP) is then used to extract image embeddings for the regions defined by the predicted boxes. These embeddings are then used as distillation targets for the region embeddings of the detection model. This approach closely follows the ViLD paper, although the present paper uses a different object detection model for the pseudo-labeling. One innovation is to add an “inter-embedding relationship matching loss”, which minimizes the difference between the similarity matrices of the CLIP (teacher) embeddings and the region embeddings of the student detector.

**Second training stage: Image-level supervision with pseudo-box labels:** In this stage, a pre-trained open-vocabulary detector is used to add box annotations to images from a classification dataset, by using the image-level (classification) labels as queries for the detector. These pseudo-labels are then used to train the classification loss of the detection models. This approach follows the Detic paper, but the present paper adds a new component called "weight transfer function". The weight transfer function conditions the model in the second stage on the embedding projection learned in the first stage, which is supposed to prevent the image-level supervision from destroying the representations learned in the first training stage.

The method is evaluated on the COCO and LVIS open-vocabulary benchmarks and compared to relevant prior work. All proposed components are ablated in experiments.



**Questions:**

1. See "Weakness 1": Provide true open-vocabulary results.
2. See "Weakness 2": Clarify MViT pretraining. If box annotations for "novel" classes were not held out for MViT training, this may inflate results. In this case, please provide results with novel-class annotations removed from all training stages.
3. Is the IRM loss computed for regions from single images, or all regions within a batch?
4. "Complimentary" --> "complementary"

**Limitations:**

Currently, the evaluation is not truly open-vocabulary, this needs to be addressed.

Societal implications are sufficiently addressed in the appendix.

**Strengths And Weaknesses:**

### Strengths
1. The paper is written very clearly. Even though the proposed methods are quite complex, they are motivated and explained well. It was a pleasure to read.

2. While the method primarily combines existing approaches from ViLD and DETIC, it it contributes important innovations that improve the synergy between these approaches (the weight transfer function and IRM loss).

### Weaknesses
1. In their current form, the presented results are **not open-vocabulary**, in contrast to the claims in the paper: the paper restricts the training data to the target vocabulary (_“Following [2, 1], we use a subset of ImageNet-21K having 997 overlapping LVIS categories and COCO captions dataset for ILS in LVIS and COCO experiments respectively.”_). This vocabulary restriction means that the method is in fact not open-vocabulary, but closed-vocabulary, since the target vocabularies (LVIS and COCO) must be known at training time for this filtering. Please either use a non-restricted training vocabulary that is not specifically tailored to the test data (e.g. all ImageNet-21k labels) for all main results, or explicitly and prominently state that the results are not open-vocabulary. For a method to be “open-vocabulary”, no part of the training pipeline should have knowledge of the label space used during testing. I emphasize that the fact that prior work (Detic) does the same is not a reason to repeat this practice. If knowledge of the test label space is required to reach good results, then these results are not "open-vocabulary", and if such restriction becomes accepted in the literature, then the term "open-vocabulary" becomes meaningless.

2. It is not clear how the MViT model used for pseudo-box prediction is trained. Does this model see any bounding box annotations during training? Is it ensured that none of these bounding box annotations include any of the "novel" classes used for evaluation?

The validity of the paper critically depends on the soundness of the evaluation. If the results hold for rigorous open-vocabulary evaluation, then the work represents and significant contribution to the field.

---

> ### Author Response · Authors · 2022-08-02
> **Response to wmjL**
>
> ## Open-vocabulary setting
>
> Thank you for the suggestion. Due to time limitation in rebuttal phase, ImageNet-21K pretraining was not possible however we are attempting it and will include in the final version.
>
> **Weak image-level supervision:**
>
> Inspired from Detic, we define our work under the **weakly-supervised open-vocabulary** setting as it uses image-level labels for expanding the detector's vocabulary. However, the complete target vocabulary set is not known (e.g., in our LVIS experiments see Table 4 in the main paper). This way, the  generalization and OVD capability of the proposed model is assessed.
>
> Prior works such as OVR-CNN and ViLD use different forms of weak-supervision to enlarge the vocabulary, such as COCO captions for pretraining or large-scale pretrained CLIP as the classifier head can have an overlap of target vocabulary, $C_{B} ∪ C_{N}$.
>
> **Generalization to cross-datasets:**
>
> Our LVIS model when evaluated on cross-datasets (OpenImages and Objects365) shows better performance compared to previous methods which indicates the generalization of our approach beyond the LVIS common, frequent and rare classes. Specifically, we ablate on the performance of cross-data evaluation by comparing overlapping categories with LVIS. In OpenImages our LVIS model attains 55.8 AP on seen classes, and 36.3 AP on unseen. In case of Objecs365, our model attains 31.3 on seen and 13.1 on unseen categories.
>
> We provide details about our experimental setup along with some additional experiment results as follows.
>
> > **COCO Split:** Our COCO model does not selectively filter any categories specific to novel categories and instead considers all COCO categories seen in the captions during the image-level supervision, consistent with OVR-CNN.
>
> > **LVIS Split:** Our LVIS model uses images from 997 overlapping ImageNet categories. However not all 337 rare categories are included in this which is a desired property for an open-vocabulary setting. Specifically, 157 rare categories do not have any overlap with image-level labels from ImageNet.
>
> We also train our LVIS model in a *stricter open-vocabulary setting* by only using LVIS base categories for image-level supervision. We achieve an overall 21.71 AP in this experiment which is similar to the model trained using image-level supervision from 997 categories (22.75 AP).
>
> ## MViT pretaining
> Though the MViT training dataset has no overlap with the evaluation data, some novel classes are exposed during MViT training. To this end, we train MViT again using the author's provided code by entirely restricting any exposure to the novel classes. The results in Tables A-2 and A-3 show that even with the exclusion of novel classes in MViT training, our approach consistently maintains a significant advantage over Detic and ViLD on COCO and LVIS datasets.
>
> |Method|AP$_r$|AP$_c$|AP$_f$|AP|
> |-|-|-|-|-|
> |ViLD (384 epochs)|16.1|20.0|**28.3**|22.5|
> |Ours (36 epochs)|**17.1**|**21.4**|26.7|**22.8**|
> Table A-2: Our LVIS results after excluding rare classes from MViT training
>
> |Method|AP$_{novel}$|AP$_{base}$|AP|
> |-|-|-|-|
> |Detic|28.4| 53.8| 47.2|
> |Ours|**36.6**|**54.0**|**49.4**|
> Table A-3: Our COCO results after excluding novel classes from MViT training
>
> ## IRM Loss
> The IRM loss is computed for regions from single images and average over all the images in a mini-batch. We will clarify this in the final version.
>
> ## Typos
> It will be corrected in the final version.

---

> > ### Comment · Reviewer_wmjL · 2022-08-04
> > **Open-vocabulary performance**
> >
> > > We also train our LVIS model in a stricter open-vocabulary setting by only using LVIS base categories for image-level supervision. We achieve an overall 21.71 AP in this experiment which is similar to the model trained using image-level supervision from 997 categories (22.75 AP).
> >
> > Thank you for providing these results. The results obtained by using only LVIS base categories should be used in the paper. That way, APr actually measures open-vocabulary performance, rather than some mixture of known and unknown-class performance.
> >
> > The same goes for the MViT pretraining: Please only show results obtained with the MViT that was not trained on novel classes.

---

> > > ### Author Response · Authors · 2022-08-05
> > > **Open-vocabulary Performance**
> > >
> > > Thank you for the suggestion and we will update our results accordingly.

---

### Official Review · Reviewer_SdFN · 2022-07-09

**Rating:** 6
**Confidence:** 4
**Soundness:** 2 fair
**Presentation:** 2 fair
**Contribution:** 3 good

**Summary:**

This paper proposes a method for open-vocabulary detection where the goal is to train an object detector able to detect objects in an image given a text description of the desired object category. The method is evaluated by its ability to detect novel classes for which no bounding box supervision is provided during train time. In more details, during training, one has access to a dataset containing bounding box annotation (e.g. COCO detection) along with a dataset that has only image level annotation (e.g. COCO captioning). The proposed approach relies on 3 core ideas: (1) a region-based knowledge distillation (RKD) approach which aims at enforcing that the features extracted from the detection vision backbone (the student) match the ones of a multimodal CLIP visual encoder (the teacher), (2) A pseudo labelling technique to extract candidate bounding boxes on the dataset containing image level annotation and (3) a method to enforce better combination of the two previous techniques by tying parameters. They report performance on COCO and LVIS and show stronger performance compared to previous approaches.

**Questions:**

**About MViT**

As discussed above, it is unclear if using MViT is coherent with the idea of trying to generalize the method to novel classes. The authors should reply to the following questions:

- Among the datasets used to train MViT, what are the ones that are annotated with bounding box annotations? Among those, are there classes that overlap with the "novel classes" of COCO or LVIS?
- Is it possible to say how much MViT alone is getting on the novel classes of COCO (it seems that it should be possible to evaluate MViT on it)?


**Method clarifications**

There are various things that are not very clear from reading the method section. Questions are listed below:

- L140: what is the difference between $\mathcal{R}$ and $\mathcal{R}(r)$?
- L143: how do you take the cosine similarity against the background vector since it consists of a vector of all zeros (how does the normalization works in that case)?
- L143: related, isnt it bad to assign boxes to background even when they could be from another class which is simply not annotated in the dataset? (e.g. a COCO images could contain objects that are not in the regular 80 COCO classes and therefore not annotated).
- L 167: How do you obtain the CLIP embedding used for RKD? Do you extract them by cropping the input images and feeding the crop to the CLIP model or do you instead extract a ROI pooled feature from a feature grid computed over the whole image?
- Figure 2: please add the total number of categories used here. From the matrix it sounds like there are 16 categories (16 * 100 images total) but I could only count 10 colors in the TSNE plot).
- L205: what is N in practice for ImageNet21K and COCO captions? In particular how do you obtain image-level category label from coco captions? I might have missed this but I could not find those details.
- Equation (4): what are the negative classes used in this equation?
- About the weight transfer function: it is unclear to me why one needs to have 2 different set of weights $W_P$ and $W_D$. The section 3.4. would benefit from better details on what exactly the architecture is and why those weights are different? The notation $f$ is used for the projection function but I found it confusing as I could not exaclty relate what $f$ precisely was with respect to the different weights $W_P$ and $W_D$. See related questions in the experiment section below.

**Question about experiment choices and details**

- Optimization details (beginning of section 4.2). Why does the method consists of multiple phases of training? Isn't it possible to train everything at once with all the additional losses?
- Related to the previous question, if everything was trained together, isnt it possible in theory to completely tie together $W_P$ and $W_D$ so that to have the exact same of weights for going from the visual feature space to the joint text-image CLIP embedding space?
- Related to the previous 2 questions, please provide details about what exactly is trained and what is frozen in the phases of training.
- Alternative to weight transfer (Table 2): it would be good to add the following two approaches:
    * $W_P=W_D$ by freezing it from the second phase of training
    * Cotraining with all the losses of L 229

**Minor requests/typos**

- Table 6 does not bold the right numbers. Table 7 is also missing a bold number (Supervised Base on AP_base)

**Limitations:**

The limitations are not discussed in the main paper. It would be good to do so: are there novel classes that suffer more than others? If yes, why? What are the important pieces of the method and how robust is the method to failure in those pieces? (e.g. the MViT component etc etc). Potential negative societal impact is also not discussed despite the method potentially being trained on large scale image and text datasets.

**Strengths And Weaknesses:**

**Strengths**

- The paper obtains good results on LVIS and COCO that outperform the state-of-the-art (Detic),
- The ideas are overall well motivated and the method is technically sound
- The ablation study is overall well conducted (see more suggestions in the Questions section)

**Weaknesses**

- An important aspect of the method seems to be that it relies on the MViT approach to obtain region proposals (used both in RKD with class agnostic proposals as well as for the pseudo labeling approach, see for Example Table 6 and Table 7 which shows the importance of MViT especially for the AP on novel classes. However MViT is trained on 3 datasets, some of which have bounding box annotation (GQA, VisualGenome and Flickr). The authors should discuss whether or not these datasets contain bounding box annotation for classes that are considered "novel" in the evaluation. If yes then I believe this would be unfair compared to previous approaches. See Questions for more details about this.

- Related to the previous point, it would be good to list in Table 3, what are the datasets that are used by each methods (e.g. what is the source image-text pairs for all methods etc etc). I encourage the authors to provide this information in the rebuttal as well. This will help to better compare against past work such as Detic.

- The description of the approach is not always clear (see Questions section for a detailed list of those questions). It is important that the authors clarify this during the rebuttal.

- Some additional experiments could be conducted (see suggestions in Questions)

=== POST REBUTTAL

After reading the rebuttal and other reviews, I am raising my score from 5 to 6. The authors have adequately addressed the important concerns I had about the novel class results. To the best of my knowledge, I believe that the results are now legit.

---

> ### Author Response · Authors · 2022-08-02
> **Response to SdFN**
>
> ## Whether MViT training Datasets Contain Box Annotations for Novel Classes?
>
> Though the MViT training dataset has no overlap with the evaluation data, some novel classes are exposed during MViT training. To this end, we train MViT again using the author's provided code by entirely restricting any exposure to the novel classes. The results in Tables A-2 and A-3 show that even with the exclusion of novel classes in MViT training, our approach consistently maintains a significant advantage over Detic and ViLD on COCO and LVIS datasets.
>
> |Method|AP$_r$|AP$_c$|AP$_f$|AP|
> |-|-|-|-|-|
> |ViLD (384 epochs)|16.1|20.0|**28.3**|22.5|
> |Ours (36 epochs)|**17.1**|**21.4**|26.7|**22.8**|
> Table A-2: Our LVIS results after excluding rare classes from MViT training
>
> |Method|AP$_{novel}$|AP$_{base}$|AP|
> |-|-|-|-|
> |Detic|28.4| 53.8| 47.2|
> |Ours|**36.6**|**54.0**|**49.4**|
> Table A-3: Our COCO results after excluding novel classes from MViT training
>
> ## Direct evaluation of MViT on COCO OVD splits
> We evaluate the performance of MViT directly on the OVD setting. From Table A-4, we observe that naively using and evaluating MViT for OVD task results in lower performance (row-2). However, class-agnostic results are relatively higher (row-3) suggesting that the classification performance of MViT is originally quite low.
>
> |Method|AP$_{novel}$|AP$_{base}$|AP|
> |-|-|-|-|
> |Supervised (Base)|1.7|53.2|39.6|
> |MViT (Class-specific)|5.6|4.1|4.5|
> |MViT (Class-agnostic)|18.5|28.0|25.5|
> |Ours|**40.3**|**54.1**|**50.5**|
> Table A-4: Evaluating MViT directly on OVD COCO
>
> MViT class-specific and class-agnostic proposals can provide better results over supervised base model for novel classes. However, it significantly lags over our final results as our other contributions like RKD (Region-based Knowledge Distillation), PIS (Pseudo-labeling for Image-level Supervision) and Weight-transfer ensure the synergy between complementary components, leading to better performance.
>
> ## Method clarifications
> **i)** $\mathcal{R}$ represents embeddings of all regions, while $\mathcal{R}(r)$ represent the region embeddings of a region $r$.
>
> **ii \& iii)** We have all zeros for the background embeddings as it makes dot product zero and its exponential to 1, avoiding pushing non-foreground bounding boxes, which may contain target/novel classes, to an arbitrary region of the embedding space.
>
> **iv)** The CLIP embeddings used for RKD are obtained by feeding the cropped image region to CLIP image encoder.
>
> **v)** In Fig. 2, the similarity matrices are computed for 17 COCO novel classes and tSNE plots are shown for 10 classes for visualization simplicity.
>
> **vi)** Here $N$ varies for each COCO caption image. The words in a caption are compared heuristically with every category name to generate a list of categories for each image ($N$), which is used as labels for PIS.
>
> **vii)** Given that we have $C_B$ total classes and $N$ positive image-level labels in an image $i$, then the negative classes would be $C_B-N$.
>
> **viii)** Our pipeline has three stages. The base model is fine-tuned for 1x schedule using RKD, where the model learns the VL projection layer $W_D$. The model is further fine-tuned to learn from PIS by adapting the VL projection layer $W_P$. During this stage, the weight transfer function explicitly conditions $W_P$ on $W_D$. Here, $W_D$ is kept frozen. This explicit conditioning is achieved as the weight transfer function transforms $W_D$ to $W_P$, via our weight transfer function $\mathcal{W_T}$.
>
> ## Optimization details
> In Table 2 (main paper), we show that naively using RKD and PIS in a single training phase is inferior to using our proposed stage-wise training setting (row 4 vs 5). This is because the RKD and PIS objectives compete with each other and our weight transfer function helps obtain complementary benefits from both.
>
> Further, we provide results of training our model with a common projection layer between RKD and PIS ($W_P$ = $W_D$), by freezing $W_D$ from second stage (Table A-5). This shows the importance of separate linear projections for individual methods and weight transfer function to mitigate competition.
>
> |Method|AP$_{novel}$|AP$_{base}$|AP|
> |-|-|-|-|
> |Ours|40.3|54.1|50.5|
> |$W_D$ $=$ $W_P$|33.5|54.1|48.7|
> Table A-5: COCO model trained with a common projection layer
>
> ## Datasets details in Table 3
> OVR-CNN uses COCO captions for pretraining. Detic uses COCO Captions and ImageNet-21k datasets while uses a pretrained RPN for proposal generation. Our approach uses COCO Captions and ImageNet-21k datasets and MViT pretrained on GQA, Visual Genome & Flickr  for proposal generation. We will clarify these details in our main paper table.
>
> ## Limitations and societal impacts
> These are discussed in supplementary material (Appendix F-H). Regarding limitation, our detailed analysis shows that the model struggles to detect small objects that have rare occurrences in the training. We will further elaborate on this in the final version.

---

> > ### Comment · Reviewer_SdFN · 2022-08-09
> > **Thanks**
> >
> > Thank you for your answers.
> >
> > I agree with Reviewer wmjL that it is absolutely crucial to update the numbers in the paper with the MViT that has not been trained on images containing bounding box annotations from the "novel classes".
> >
> >
> > By the way I think that question was omitted in your response:
> >
> > > Related to the previous 2 questions, please provide details about what exactly is trained and what is frozen in the phases of training.
> >
> > It would be great if you could provide an answer to this and clarify in the paper.
> >
> > Thank you.

---

> > > ### Author Response · Authors · 2022-08-09
> > > **Response to SdFN**
> > >
> > > Thank you for the suggestion. We will update our results accordingly. As noted from our updated results, the MViT trained after excluding novel class box annotations performs well and our final model still provides performance gain compared to existing state of the art approaches, ViLD (ICLR 2022) and Detic (ECCV 2022).
> > >
> > > Regarding the details about what exactly is trained and what is frozen in the phases of training, we follow a three stage training strategy. Specifically, in stage-1, the base model is first trained for 1x schedule using only base class annotations using a simple linear projection layer. In stage-2, the model is fine-tuned for another 1x schedule using Region-based Knowledge Distillation (RKD), where it adapts the projection layer in stage-1 to align the local region representations to the language representations. We refer to this adapted projection layer as $W_D$. Finally in stage-3, the model is further fine-tuned to learn from weak image-level supervision with pseudo-box labels (PIS) by learning the VL projection layer $W_P$. During this stage, the weight transfer function explicitly conditions $W_P$ on the mapping learned from distillation, $W_D$. Here VL projection layer $W_D$ is kept frozen. All the losses in L229 are used during the last stage of the training. The above details were also provided under "Method Clarifications" (point viii) of our previous response.
> > >
> > > We will further clarify above details in section 4.2 (implementation details) of the paper.

---

### Official Review · Reviewer_LKon · 2022-07-11

**Rating:** 6
**Confidence:** 4
**Soundness:** 4 excellent
**Presentation:** 3 good
**Contribution:** 3 good

**Summary:**

This paper proposed a training pipeline for better dealing with open-vocabulary object detection. Specifically, it incorporates CLIP as a pre-trained vision-language embedding to significantly expand the potential object detection vocabulary and mitigated performance degradation due to the gap between image-centric and object-centric representation. In this paper three important components are proposed to enhance the system, including (1) region-based knowledge distillation to match the CLIP embedding for image patch (instead of full image) with text, thus the replaced text classification head can better fit the region detection; (2) image-level supervision with pseudo box labels which benefit from a recent multi-modal ViT (MViT) class-agnostic object detector; (3) a weight transfer function which combines (1) and (2) more effectively in a trainable approach. The overall system is validated on multiple benchmarks and achieves state-of-the-art performance, especially for novel object classes.

**Questions:**

1. Although the authors perform detailed ablation studies to validate each component in their system, one critical aspect is not covered which is the performance of MViT on "novel" object proposal/detection. Specifically, since the region proposal is generated from MViT for both class-agnostic and class-specific approaches, one possibility is that the pre-training of MViT may have some leaked information to the student model. The authors did mention that "we ensure that MViT pretraining dataset has no overlap with any of the evaluation datasets in our work." but I'm curious about details about the exclusion -- does it mean the eval images are never included in the pre-training or even the novel object classes are not exposed? Probably an even more careful ablation study would be replacing RPN in the baseline object detection with MViT so the effect of the enhanced proposal can be analyzed. Essentially I'm feeling that the quality of the system is heavily dependent on the $\mathcal{Q}_{pseudo}$ but this paper did not provide sufficient ablations about it.

2. Regarding the weight transfer function, I'm curious how this is trained -- is it being trained in an end-to-end fashion (I presume) together with the full system using loss defined in L229? Does it mean $W_P$ in the weight transfer function is now replaced by $W_\tau$ and $W_D$? Probably it's better to explicitly show this relation in the equation below L229.

**Limitations:**

This author does not provide explicit explanations of limitations and societal impacts but I presume this system has similar issues with other systems (e.g. potential societal bias when trained with large-scale automatic collected datasets).

**Strengths And Weaknesses:**

Strengths:

- The assumption for the performance degradation from CLIP to object detection is reasonable and the authors propose a solid solution to mitigate the gap. The image-centric and object-centric nature may lead to incompatibility of the visual encoder and textual embedding, thus introducing the $L1$ and $L2$ regularization objectives is intuitively helpful to bridge the gap.
- The weight transfer function seems interesting and shows great potential for merging two components in a more harmonized way. The performance boost is quite significant and similar ideas may inspire other ensembling/multi-component fusion systems.
- The details of the training and evaluation are provided and the paper is easy to follow.

Weaknesses:

- The model did demonstrate significant improvements over other approaches, although it doesn't seem clear to me how much of the distillation from MViT is involved in these improvements. See my Questions.

---

> ### Author Response · Authors · 2022-08-02
> **Response to LKon**
>
> The main question posed by Reviewer LKon is to assess how much the distillation from MViT contributes to our performance improvement. To answer this question, we report below ablation studies as well as new experimental results on refined data splits. Our conclusion is that although MViT is an important component of our approach, other contributions of this work including RKD (Region-based Knowledge Distillation), PIS (Pseudo-labeling for Image-level Supervision) and Weight Transfer are important towards developing an integrated framework for OVD. An important contribution of our approach is a careful design for unifying the advantages of generic object detector (MViT), knowledge distillation and pseudo-labeling.  This is also highlighted by Reviewers wmjL, SdFN and dzUw,  e.g., by mentioning
> >It is the first paper in attempt to combine the advantages of both knowledge distillation and self-training.
>
> ## Ablation study replacing RPN proposals with proposals from MViT
>
> We refer to the Table A-1 where we use max-size strategy from Detic with RPN proposals (rows 2-3) in comparison with MViT proposals (row-4). We show that using MViT proposals provides improvements, indicating the suitability of proposals from MViT. However, an equally important question is *how to effectively use these proposals from MViT*, which we solve using our proposed pseudo-labeling strategy, $\mathcal{Q}_{\text{pseudo}}$ (row-5). In addition to MViT proposals, our other proposed components including RKD and Weight Transfer proves to be equally valuable (row-6).
>
> | Method| AP$_{novel}$ | AP$_{base}$ |AP|
> |-|-|-|-|
> |1: Supervised (Base)|1.7|53.2|39.6|
> |2: Detic with RPN (Max-Score loss)|15.9|48.2|39.7|
> |3: Detic with RPN (Max-Size loss)|25.9|51.1|44.5|
> |4: Detic with MViT (Max-Size loss)|28.9|50.7|45.0|
> |5: Pseudo-box on MViT (Ours)|34.2|52.0|47.4|
> |6: Ours (RKD + PIS + Weight-transfer)|**40.3**|**54.1**|**50.5**|
> Table A-1: An ablation study on replacing RPN in Detic with MViT on COCO dataset
>
> ## Exclusion of novel classes in MViT training
>
> Though the MViT training dataset has no overlap with the evaluation data, some novel classes are exposed during MViT training. To this end, we perform the following additional experiments.
>
> 1. We train MViT again using the author's provided code by entirely restricting any exposure to the novel/rare classes. The results in Tables A-2 and A-3 show that even with the exclusion of novel classes in MViT training, our approach consistently maintains a significant advantage over Detic and ViLD on COCO and LVIS datasets.
>
> |Method|AP$_r$|AP$_c$|AP$_f$|AP|
> |-|-|-|-|-|
> |ViLD (384 epochs)|16.1|20.0|**28.3**|22.5|
> |Ours (36 epochs)|**17.1**|**21.4**|26.7|**22.8**|
> Table A-2: Our LVIS results after excluding rare classes from MViT training
>
> |Method|AP$_{novel}$|AP$_{base}$|AP|
> |-|-|-|-|
> |Detic|28.4| 53.8| 47.2|
> |Ours|**36.6**|**54.0**|**49.4**|
> Table A-3: Our COCO results after excluding novel classes from MViT training
>
> 2. We evaluate the performance of MViT directly on the OVD setting. From Table A-4, we observe that naively using and evaluating MViT for OVD task results in lower performance (row-2). However, class-agnostic results are relatively higher (row-3) suggesting that the classification performance of MViT is originally very low.
>
> |Method|AP$_{novel}$|AP$_{base}$|AP|
> |-|-|-|-|
> |Supervised (Base)|1.7|53.2|39.6|
> |MViT (Class-specific)|5.6|4.1|4.5|
> |MViT (Class-agnostic)|18.5|28.0|25.5|
> |Ours|**40.3**|**54.1**|50.5|
> Table A4: Evaluating MViT directly on OVD COCO
>
> MViT class-specific and class-agnostic proposals can provide better results over supervised base model for novel classes. However, it significantly lags over our final results as our other contributions like RKD, PIS and Weight-transfer ensure the synergy between complementary components, leading to best performance.
>
> ## Training using weight transfer function
> Our proposed pipeline has three stages. The base model is fine-tuned for 1x schedule using RKD, where the model learns the VL projection layer $W_D$. The model is further fine-tuned to learn from PIS by adapting the VL projection layer $W_P$. During this stage, the weight transfer function explicitly conditions $W_P$ on $W_D$. All the losses in L229 are used during this stage. This explicit conditioning is achieved as the weight transfer function transforms $W_D$ to $W_P$, via our weight transfer function $\mathcal{W_T}$. Further, in Table 2 (main paper), we show that naively using RKD and PIS in a single training phase is inferior to using our proposed stage-wise training setting (row 4 vs 5). This is because the RKD and PIS objectives compete with each other and our proposed weight transfer function helps obtain complementary benefits from both.
>
> ## Limitations and societal impacts
> These are discussed in supplementary material (Appendix F-H).

---

> > ### Comment · Reviewer_LKon · 2022-08-09
> > **Thanks**
> >
> > Thanks for the response, and the additional experiments. I've also read the comments from other reviewers (SdFN and wmjL) and agree with adding/replacing experiment results about MViT. I'll keep my current rating at this moment.

---

### Author Response · Authors · 2022-08-02
**Thank you for the valuable feedback**

We sincerely thank all the reviewers (LKon, SdFN, wmjL, dzUw) for their detailed and constructive feedback. All reviewers appreciate the technical soundness and innovations of the proposed components in our open-vocabulary object detection (OVD) pipeline. Specifically, the proposed approach used for combining different components (dzUw) and improving their synergies (wmjL) may inspire other ensembling/multi-component fusion systems (LKon). The proposed approach results in better performance on OVD benchmark datasets (LKon, SdFN) and the ablation studies have been noted to be thorough and well conducted (SdFN, dzUw). The reviewers also appreciate the clear writing and presentation of the paper (LKon, wmjL, dzUw).

Our code and pretrained models will be publicly released.

---

### Author Response · Authors · 2022-08-09
**Thank you for the feedback and support**

We thank all the reviewers for going through our response and providing support. As suggested by the reviewers, we will update the numbers which are obtained after exclusion of all novel/rare classes. Our response shows that the benefit of our approach in comparison to state-of-the-art methods ViLD (ICLR'22) and Detic (ECCV'22) holds under the new setting. We believe that all the queries have been answered and we welcome any further feedback from the reviewers.

---

### Meta-Review · Area_Chair_TQ3h · 2022-08-27

**Recommendation:** Accept
**Confidence:** Certain

**Metareview:**

The paper receives overall positive ratings after rebuttal. The major concern before rebuttal is that the benefits and limitations from using MViT are unclear. The rebuttal has addressed most concerns from reviewers. AC encourages authors to make the final revision with review comments.

**Award:**

No

---

### Decision · Program_Chairs · 2022-09-14

Accept